# FK506-Binding Protein 11 Is a Novel Plasma Cell-Specific Antibody Folding Catalyst with Increased Expression in Idiopathic Pulmonary Fibrosis

**DOI:** 10.3390/cells11081341

**Published:** 2022-04-14

**Authors:** Stefan Preisendörfer, Yoshihiro Ishikawa, Elisabeth Hennen, Stephan Winklmeier, Jonas C. Schupp, Larissa Knüppel, Isis E. Fernandez, Leonhard Binzenhöfer, Andrew Flatley, Brenda M. Juan-Guardela, Clemens Ruppert, Andreas Guenther, Marion Frankenberger, Rudolf A. Hatz, Nikolaus Kneidinger, Jürgen Behr, Regina Feederle, Aloys Schepers, Anne Hilgendorff, Naftali Kaminski, Edgar Meinl, Hans Peter Bächinger, Oliver Eickelberg, Claudia A. Staab-Weijnitz

**Affiliations:** 1Institute of Lung Health and Immunity and Comprehensive Pneumology Center with the CPC-M bioArchive, Member of the German Center of Lung Research (DZL), Helmholtz-Zentrum München, 81377 Munich, Germany; preisendoerfer@dhm.mhn.de (S.P.); elisabeth.hennen@helmholtz-muenchen.de (E.H.); larissa.knueppel@secarna.com (L.K.); isis.fernandez@helmholtz-muenchen.de (I.E.F.); leonhard.binzenhoefer@med.uni-muenchen.de (L.B.); frankenberger@helmholtz-muenchen.de (M.F.); anne.hilgendorff@helmholtz-muenchen.de (A.H.); eickelbergo@upmc.edu (O.E.); 2Department of Biochemistry and Molecular Biology, Oregon Health & Science University, Portland, OR 97239, USA; yoshihiro.ishikawa@ucsf.edu (Y.I.); hanspeter.bachinger@gmail.com (H.P.B.); 3Institute of Clinical Neuroimmunology, Biomedical Center and LMU Klinikum, Ludwig-Maximilians-Universität München, 81377 Munich, Germany; stephan.winklmeier@med.uni-muenchen.de (S.W.); edgar.meinl@med.uni-muenchen.de (E.M.); 4Pulmonary, Critical Care and Sleep Medicine, Yale School of Medicine, New Haven, CT 06520, USA; jonas.schupp@yale.edu (J.C.S.); brendajuan@usf.edu (B.M.J.-G.); naftali.kaminski@yale.edu (N.K.); 5Department of Respiratory Medicine, Hannover Medical School, Biomedical Research in End-Stage and Obstructive Lung Disease Hannover, Member of the German Center for Lung Research (DZL), 30625 Hannover, Germany; 6Department of Medicine V, LMU Klinikum, Ludwig-Maximilians-Universität München, Member of the German Center of Lung Research (DZL), 81377 Munich, Germany; nikolaus.kneidinger@med.uni-muenchen.de (N.K.); juergen.behr@med.uni-muenchen.de (J.B.); 7Monoclonal Antibody Core Facility, Institute for Diabetes and Obesity, Helmholtz-Zentrum München, 85764 Neuherberg, Germany; andrew.flatley@helmholtz-muenchen.de (A.F.); regina.feederle@helmholtz-muenchen.de (R.F.); schepers@helmholtz-munich.de (A.S.); 8Department of Internal Medicine, Medizinische Klinik II, Member of the German Center of Lung Research (DZL), 35392 Giessen, Germany; clemens.ruppert@innere.med.uni-giessen.de (C.R.); andreas.guenther@innere.med.uni-giessen.de (A.G.); 9Thoraxchirurgisches Zentrum, Klinik für Allgemeine-, Viszeral-, Transplantations-, Gefäß- und Thoraxchirurgie, LMU Klinikum, Ludwig-Maximilians-Universität München, 81377 Munich, Germany; rudolf.hatz@med.uni-muenchen.de; 10Asklepios Fachkliniken München-Gauting, 82131 Gauting, Germany

**Keywords:** antibody folding, immunophilin, lung fibrosis, interstitial lung disease, ER stress, tacrolimus, FK506-binding protein, peptidyl-prolyl isomerase

## Abstract

Antibodies are central effectors of the adaptive immune response, widespread used therapeutics, but also potentially disease-causing biomolecules. Antibody folding catalysts in the plasma cell are incompletely defined. Idiopathic pulmonary fibrosis (IPF) is a fatal chronic lung disease with increasingly recognized autoimmune features. We found elevated expression of FK506-binding protein 11 (*FKBP11*) in IPF lungs where FKBP11 specifically localized to antibody-producing plasma cells. Suggesting a general role in plasma cells, plasma cell-specific *FKBP11* expression was equally observed in lymphatic tissues, and in vitro B cell to plasma cell differentiation was accompanied by induction of *FKBP11* expression. Recombinant human FKBP11 was able to refold IgG antibody in vitro and inhibited by FK506, strongly supporting a function as antibody peptidyl-prolyl *cis-trans* isomerase. Induction of ER stress in cell lines demonstrated induction of *FKBP11* in the context of the unfolded protein response in an X-box-binding protein 1 (XBP1)-dependent manner. While deficiency of FKBP11 increased susceptibility to ER stress-mediated cell death in an alveolar epithelial cell line, FKBP11 knockdown in an antibody-producing hybridoma cell line neither induced cell death nor decreased expression or secretion of IgG antibody. Similarly, antibody secretion by the same hybridoma cell line was not affected by knockdown of the established antibody peptidyl-prolyl isomerase cyclophilin B. The results are consistent with FKBP11 as a novel XBP1-regulated antibody peptidyl-prolyl *cis-trans* isomerase and indicate significant redundancy in the ER-resident folding machinery of antibody-producing hybridoma cells.

## 1. Introduction

Plasma cell-secreted antibodies are ultimate effectors of the adaptive immune response and fundamentally important for the neutralization of pathogenic virus and bacteria [1]. Their activity relies on high-affinity binding to their cognate antigen, which requires correct folding into the functional three-dimensional structure in the endoplasmic reticulum (ER). This process is strictly dependent on an ER-resident protein folding machinery consisting of molecular chaperones, peptidyl-prolyl *cis*-*trans* isomerases (PPIases), disulfide isomerases, and glycosyl transferases [2,3]. During B-cell to plasma cell differentiation, components of this machinery are increased in the course of the so-called unfolded protein response (UPR) to accommodate the increasing burden of incoming nascent immunoglobulins and avoid accumulation of misfolded proteins in the ER [4,5]. X-box-binding protein 1 (XBP1) is a central transcription factor orchestrating the UPR by induction of chaperones and other protein folding catalysts during late-stage plasma cell differentiation [6,7,8].

Peptidyl-prolyl *cis*-*trans* isomerization often represents a rate-limiting step in protein folding. Prolines make up 5–10% of an antibody’s primary sequence and facilitate the formation of turns connecting the β-strands in each immunoglobulin (Ig) fold. Two protein families collectively termed immunophilins exert PPIase activity in the ER, namely the FK506-binding proteins (FKBPs) and the cyclophilins [9]. Until now, limited evidence suggests that two PPIases may directly participate in antibody folding including peptidyl-prolyl *cis-trans* isomerase B (PPIB, also termed cyclophilin B, CypB) [10,11,12,13] and FKBP1A (also termed FKBP12) [14]. The latter, however, typically resides in the cytosol [15] and thus is unlikely to play a role in antibody folding in vivo.

*FKBP11* (also termed FKBP19) was first described as predominantly expressed in secretory tissues including pancreas [16] and recent studies have suggested a role in beta cell survival under conditions of ER stress [17,18,19]. In addition, FKBP11 appears to play a role in osteoblasts and bone formation where it has been observed to associate with interferon-inducible transmembrane protein 5 (IFITM5) [20,21,22]. Other studies have proposed FKBP11 as a prognostic marker for hepatocellular carcinoma [23,24]. Many of the studies mentioned above point towards a role in the context of ER stress. Interestingly, gene expression profiling studies show upregulation of *FKBP11* during differentiation from B cells to antibody-secreting plasma cells and indicate that, in this context, *FKBP11* expression may be regulated by XBP1 [25,26]. Finally, overexpression of *FKBP11* in murine splenic B cells has been reported to increase initiation of plasma cell differentiation [27]. These observations suggest that FKBP11 may be involved in the plasma cell UPR and/or antibody folding but, to date, function and substrate of this protein have remained largely obscure.

Importantly, a profound understanding of mechanisms of plasma cell differentiation and antibody production may be beneficial for targeting autoimmune diseases where autoantibodies play a crucial pathogenetic role. Current treatment options include antibodies specifically targeting the B-cell lineage, the most prominent example being rituximab, a monoclonal antibody directed towards CD20, a pan B-cell surface protein [28], but B cell depletion shows a limited impact on the plasma cell population [29]. Other therapeutic options than complete B-cell depletion have proven successful in murine models of autoimmune disease, e.g., extracellular cleavage of autoantibodies [30]. Clearly, elucidation of the specific antibody folding machinery in plasma cells may contribute to the development of further therapeutic strategies. Notably, *FKBP11* is overexpressed in B-cells of patients suffering from Systemic Lupus Erythematosus, a chronic inflammatory autoimmune disease [27]. 

Idiopathic pulmonary fibrosis (IPF) is a fatal disease with median survival rates ranging from 3–5 years, increasing incidence worldwide, and limited treatment options [31,32]. Even if it is currently believed that IPF pathogenesis originates from micro-injuries to the airway and alveolar epithelium, the etiology of IPF is incompletely understood [33,34]. In contrast, it is well-known that many autoimmune diseases manifest in lung fibrosis [35] and studies in experimental mouse models have shown that autoantibodies can cause and promote lung fibrosis [36,37,38,39], while B-cell depletion is protective [40,41,42]. Notably, evidence is accumulating which also argues for significant autoimmune features in IPF: A recent proteomic study has revealed increased numbers of MZB1-positive plasma cells and higher IgG levels in tissue of patients suffering from various interstitial lung diseases (ILD) including IPF [43]. In blood of IPF patients, higher levels of circulating plasmablasts, several soluble factors that promote B cell growth and differentiation, and various autoantibodies towards lung antigens have been reported [44,45]. Finally, distinct (human leukocyte antigen (HLA) class II alleles are overrepresented in patients with IPF, thus equally supporting a role of autoimmunity in IPF aetiology [46,47]. 

Here, we report that FKBP11 is a novel antibody folding catalyst, levels of which are strongly increased in IPF, specifically produced by human plasma cells, and induced by the UPR in an XBP1-dependent manner.

Some of the results of these studies have been previously reported in the form of conference abstracts [48,49].

## 2. Materials and Methods

### 2.1. Patient Samples

Patient lung tissue and blood samples were obtained from the CPC-M bioArchive at the Comprehensive Pneumology Center (CPC) and from the UGMLC Giessen Biobank, member of the DZL platform biobanking. For samples from the CPC-M BioArchive and the biobank of the Institute of Clinical Neuroimmunology, the study was approved by the local Ethics Committee of Ludwig-Maximilians University of Munich, Germany (333-10, 382-10, and 163-16, respectively). Biomaterial collection by the UGMLC/DZL biobank was approved by votes from the Ethics Committee of the Justus-Liebig-University School of Medicine (111/08, and 58/15). Informed consent was obtained from each subject. The tissue microarray (Multi-normal human tissues, 96 samples, 35 organs/sites from three individuals) was from Abcam (ab178228) and contained only healthy specimens except for tonsils which were inflamed (tonsillitis).

### 2.2. Gene Expression Data

Gene expression data for *FKBP* genes in normal histology control (*n* = 43) and IPF lungs (*n* = 99) was extracted from microarray data generated by us on lung samples obtained from the National Lung, Heart, and Blood Institute-funded Tissue Resource Consortium (NLHBI LTRC), as described previously [50,51]. Gene expression microarray data (Agilent Technologies, Santa Clara, CA, USA) are available under accession number GSE47460 in the data set repository Gene Expression Omnibus (GEO). Significance was calculated using *t* statistics, and multiple testing was controlled by the false discovery rate method at 5% [52].

### 2.3. Cell Culture, Induction of ER Stress and Transfection

For details on culture, treatment, and transfection of A549 and Raji cells, see the online supplement. The mouse myeloma cell line P3X63-Ag8.653 (AG8) and mouse/rat hybridoma cell lines were maintained in the same manner as described for Raji cells (see online supplement). 

Delivery of scrambled, anti-FKBP11, and anti-PPIB siRNA into the hybridoma cell line H3 was achieved by electroporation largely as described [53]. Briefly, hybridoma cells at a culture density between 3 × 10^5^–5 × 10^5^ cells/mL were spun down at 300× *g* and resuspended in fresh RPMI-1640 medium (Life Technologies, Carlsbad, CA, USA) without supplements at a concentration of 6 × 10^7^ cells/mL. After addition of anti-FKBP11 siRNA (s82617, Life Technologies, targeting both rat and mouse FKBP11), anti-PPIB siRNA (s72031), or negative control siRNA No. 1 (Life Technologies) at a final concentration of 3 µM, 200 µL of the mixture were transferred to a 96-well electroporation plate (Bio-Rad, München, Germany). Electroporation was then carried out with a Gene Pulser (Bio-Rad, München) by application of a single exponential decay pulse at a capacity setting of 950 µF and a voltage of 300 V. Subsequently, cells were transferred to prewarmed full medium, incubated at 37 °C and cells and supernatant harvested after 48 h.

### 2.4. Plasma Cell Differentiation

PBMCs were isolated from blood of healthy volunteers using density gradient centrifugation (described in detail in the online supplement). PBMCs were transferred to pre-warmed RPMI medium (Thermo Fisher Scientific, Schwerte, Germany) supplemented with 10% FBS, L-glutamine, sodium bicarbonate and non-essential amino acids and once washed. Afterwards, PBMCs were seeded in a cell culture dish at a concentration of 1 × 10^6^ cells/mL and stimulated by 1000 U/mL of recombinant IL-2 (Roche) and 2.5 µg/mL TLR 7 + 8 ligand R848 (InvivoGen, San Diego, CA, USA), an established procedure for the induction of plasma cell transdifferentiation [54,55]. As a control, same volumes of diluents were added to PBMCs. After 7 days of incubation at 37 °C, supernatants were collected to determine IgG concentrations, cytospins were prepared, and RNA and protein was extracted from remaining cells.

For flow cytometric analysis of cell populations following stimulation, activated and differentiated cells were stained using anti-human CD3-Alexa Fluor 700 (OKT3; eBioscience, San Diego, CA, USA), CD19-APC/Fire 750 (HIB19; BioLegend, San Diego, CA, USA), CD27-Brilliant Violet 605 (O323; BioLegend), CD38-eFluor 450 (HB7; eBioscience), FcR blocking reagent (Miltenyi Biotec, Bergisch Gladbach, Germany) and TO-PRO-3 (Invitrogen, Eugene, OR, USA). Subsequently, cells were pre-gated on live and singlet cells and further gated on CD3^−^ and CD19^+^ B cells as described in Winklmeier et al. [56]. Plasmablasts (CD19^+^CD27^high^CD38^high^) and non-plasmablast B cells (CD19^+^CD27^low^CD38^low^) were bulk sorted using FACSAria Fusion (BD, Franklin Lakes, NJ, USA. In some experiments, CD3^+^ T cells were removed before the FACS sorting using magnetic beads (EasySep™ Human CD3 Positive Selection Kit II, STEMCELL Technologies, Vancouver, Canada). Cells were lysed in RLT Plus buffer containing DTT according RNeasy Plus Mini Kit (QIAGEN, Hilden, Germany). Analysis of flow cytometry data was performed with FlowJo (V10.6.1, BD).

### 2.5. RNA Isolation and Real-Time Quantitative Reverse-Transcriptase PCR (qRT-PCR) Analysis

For details on isolation of RNA from cell cultures, reverse transcription and PCR analysis, see the online supplement.

### 2.6. Protein Isolation and Western Blot Analysis

For details on protein isolation and Western Blot analysis, see the online supplement.

### 2.7. Flow Cytometry Analysis

To assess the number of circulating plasma cells in blood of IPF patients in comparison with healthy donors, plasma cells from whole blood of IPF patients and healthy donors were sorted and identified as CD20^−^/CD3^−^/CD27^+^/CD38^+^ cells. First, venous blood from IPF patients or healthy individuals was collected in EDTA-coated vacutainer tubes (Sarstedt, Nümbrecht, Germany). For each staining, 100 µL of blood was incubated with antibodies (listed in Appendix A) used to gate for plasma cells (see Appendix A for gating strategy) for 20 min at 4 °C protected from light. In parallel, blood from the same specimen was stained with appropriate isotype controls. Next, lysis of erythrocytes was performed with a Coulter Q-Prep working station (Beckman Coulter, München, Germany) followed by data acquisition in a BD LSRII flow cytometer (Becton Dickinson, Heidelberg, Germany). FlowJo software (TreeStart Inc., Ashland, OR, USA) was used for data analysis. Data was presented as ratio CD20^−^/CD3^−^/CD27^+^/CD38^+^ cells of live cells. Negative thresholds for gating were set according to isotype-labelled controls. 

### 2.8. Immunofluorescent Stainings

For details on immunofluorescent stainings of tissue sections and cytospins, please refer to the online supplement. 

### 2.9. Unfolding and Refolding of Immunoglobulin G

Antibody refolding was performed essentially as described [14,57]. A monoclonal mouse anti-fibrillin-1 antibody [58] was denatured using 3 M guanidinium chloride pH 7.0, 0.1 M Tris, 0.005 M EDTA) for 24 h at an antibody concentration of 80 µg/mL at 4 °C. For initiation of refolding, the denatured antibody solution was diluted 10-fold under manual shaking in PBS containing a PPIase or a control protein (RNAse 45 µM, Sigma-Aldrich) at 10 °C, resulting in an antibody concentration of 8 µg/mL. Recombinant PPIases used were either PPIB (5 µM, positive control) or FKBP11 (45 µM) [59]. At given time points, aliquots of the refolding mixture were withdrawn and diluted 12-fold under vigorous stirring in a trypsin solution (300 U/mL trypsin, 5% milk, PBS) and kept on ice to stop further refolding. Final antibody concentrations were 0.66 µg/mL. After completion of the final time point, the amount of correctly refolded antibody was determined by ELISA (see below).

For experiments involving inhibition of PPIase activity, FKBP11 was preincubated for one hour with either FK506 (180 µM, Sigma-Aldrich, St. Louis, MO, USA) or DMSO (Sigma-Aldrich) as a negative control at +10 °C. Subsequently, antibody refolding was performed as described above.

### 2.10. ELISA

To assess the rate of correctly refolded IgG, a high binding ELISA plate was coated with recombinant fibrillin (fragment rf11) [58] overnight. After washing one time with TBS-Tween (0.025% Tween 20), the coated wells were blocked with 5% milk in PBS for one hour. Then, aliquots from the refolding experiment were incubated for one hour. The plate was washed once, followed by one hour of incubation with an HRP-linked anti-mouse antibody (Bio-Rad, Hercules). After rinsing the plate 5 times, 3,3′,5,5′-Tetramethylbenzidine (TMB) substrate (Sigma-Aldrich) was incubated for 20 min and the signal was read at 650 nm. In the graphs, values shown represent the optical density measured at 650 nm minus blank values (derived from incubation with trypsin solution without primary antibody). 

For determination of yield and functionality of antibodies secreted from scr siRNA, FKBP11 siRNA or PPIB siRNA-transfected hybridoma H3 cells, ELISA plates were coated with a mixture of anti-κ LC (TIB172, ATCC) and anti-λ LC (mAb LA1B12 [60], both 5 µg/mL in 0.2 M carbonate buffer pH 9.5), the cognate antigen-GST fusion protein, or the untagged cognate antigen (each 10 µg/mL in 0.2 M carbonate buffer pH 9.5, in-house generated) overnight at 4 °C. After washing one time with PBS, the wells were blocked with PBS/2% fetal calf serum for 15 min. After blocking, a serial 2-fold dilution of supernatants (starting dilution 1:10) from siRNA-transfected hybridoma cells were added for 30 min. The plate was washed once, followed by 30 min of incubation with an HRP-linked mouse-anti-rat IgG2a, which is the IgG subclass of the antibody produced by this hybridoma cell line. After rinsing the plate 5 times with PBS, 3,3′,5,5′-Tetramethylbenzidine (TMB) substrate (Thermo Scientific/Pierce) was added, followed by incubation for 5 minutes and monitoring of absorbance at 650 nm.

## 3. Results

### 3.1. FKBP11 Expression Is Increased in IPF Lungs

As we had previously observed increased *FKBP10* expression in IPF [61], we set out to assess expression of other members of the *FKBP* family in IPF in microarray data of 99 IPF samples and 43 normal histology control samples [50,51]. With a false discovery rate of 5% [52], we found four more *FKBP*s to be significantly differentially expressed, namely *FKBP11*, *FKBP1A*, *FKBP5*, and *FKBP6*. *FKBP11* and *FKBP5*, but not *FKBP1A* and *FKBP6*, were expressed at comparably high abundance and altered more than two-fold, namely *FKBP11* with a Fold Change of +2.2 and *FKBP5* with a Fold Change of -3.4 (Figure 1A). Focusing on these two FKBPs, we found that, in contrast to transcript levels, FKBP5 protein levels were not decreased (Appendix A). However, immunoblot analysis demonstrated increased FKBP11 protein in IPF relative to healthy donor lung samples (Figure 1B,C).

Taking advantage of data available in the NLHBI LTRC data set [62], we correlated *FKBP11* expression with demographic and clinical parameters including age, sex, lung function, several readouts of quality of life and physical fitness as well as smoking history. We observed a significant (*p* = 0.02) but weak negative correlation with FVC (% of predicted) and a similar relationship with DLCO which, however, just failed significance (*p* = 0.06; Appendix A).

### 3.2. FKBP11 Localizes Mainly to CD27^+^/CD38^+^/CD138^+^/CD20^−^/CD45^−^ Plasma Cells

We initially assessed whether *FKBP11*, like *FKBP10*, was expressed in primary human lung fibroblasts (phLF) and contributed to collagen synthesis and secretion [61]. Even if we detected FKBP11 in the microsomal fraction of phLF, in agreement with ER residence (Appendix A), *FKBP11* was significantly less expressed in phLF than *FKBP10* (Appendix A) and siRNA-mediated downregulation of FKBP11 did neither affect myofibroblast differentiation (as assessed by protein levels of α-smooth muscle actin, Appendix A) nor collagen secretion from these cells (Appendix A).

Immunofluorescent stainings of IPF lung tissue sections for different markers of the hematopoietic lineage revealed that FKBP11 specifically localizes to CD27^+^/CD38^+^/CD138^+^/CD20^−^/CD45^−^ plasma cells in IPF lungs (Figure 2A,B). Our results further supported expression of *FKBP11* predominantly by IgG-producing plasma cells, but we also observed IgA/FKBP11 double-positive cells (Figure 2B). Elevated cell counts of FKBP11^+^/CD38^+^ plasma cells (Figure 2C) confirmed overexpression of *FBKP11* in IPF lungs and suggested that upregulation of *FKBP11* in IPF lung tissue is mostly due to increased prevalence of tissue-resident plasma cells. In agreement, *FKBP11* expression very strongly correlated with *MZB1*, an established marker of plasma cells [43], in IPF lung tissue, and not or only weakly with typical fibrotic markers such as *ACTA2*, *COL1A2*, and *COL3A1* (Appendix A). In contrast, proportions of CD38^+^/CD27^+^ cells in fresh whole blood samples analyzed by FACS analysis did not significantly differ between IPF and donor samples (Figure 2D, for gating strategy refer to Appendix A). We took advantage of a tissue microarray and observed that *FKBP11* was also expressed by CD38^+^ plasma cells in primary and secondary lymphatic organs including thymus, spleen, tonsils and small intestine, suggesting that *FKBP11* expression is a common property of plasma cells (Figure 2E). We also found FKBP11^+^/CD38^−^ cells in pancreatic and gastric glands (Appendix A), in line with an important function of FKBP11 in secretory cells in other tissues. In contrast, on the same tissue microarray we did not detect neither FKBP11 nor CD38 in healthy human tissues such as muscle and lung (Appendix A).

### 3.3. FKBP11 Is Upregulated during Plasma Cell Transdifferentiation

Isolation and subsequent 7-day treatment of peripheral blood mononuclear cells (PBMCs) from three independent healthy donors with a combination of IL2 and R848 led to transdifferentiation of memory B cells to IgG producing plasma cells (Figure 3A,B), as demonstrated before [43,54,55]. Upon treatment, immunofluorescent stainings of cytospins showed more IgG^+^ cells than control (Figure 3A). In addition, transcript levels of PR domain zinc finger protein 1 (*PRDM1*, also termed BLIMP-1), a marker of B-cell activation [63], were significantly increased (Figure 3B). At the same time, B cell to plasma cell transdifferentiation was accompanied by an upregulation of FKBP11 on both transcript (Figure 3C) and protein level (Figure 3D). Upregulation of the ER chaperone HSPA5 (also termed BiP or GRP78) confirmed induction of the unfolded protein response (UPR) during differentiation to plasma cells (Figure 3D). In order to verify that upregulation of *FKBP11* indeed was due to B cell to plasmablast differentiation, we used flow cytometry to select the CD3^−^CD19^+^ B cell population from IL2/R848-activated cells, sorted those into plasmablasts (CD27^high^ CD38^high^) and non-plasmablast B cells (CD27^low^ CD38^low^), and directly compared *FKBP11* and *PRDM1* gene expression in these B cell populations (Figure 3E,F). Similar to *PRDM1*, *FKBP11* was highly enriched in the CD27^high^CD38^high^ plasmablast population.

Finally, single cell-RNA-Seq analysis of healthy mouse lungs extracted from Angelidis et al [64] confirms plasma cells as the major source of *Fkbp11* in the lung (Appendix A), similar to *Mzb1*, which was previously reported by us as a plasma cell-specific protein upregulated in human lung fibrosis [43]. This data also showed marginal expression of *Fkbp11* in interstitial fibroblasts and alveolar epithelial cells, which is in agreement with our qRT-PCR and immunoblot results in phLF (Appendix A) and A549 (Figure 4). Notably, *PPIB*, typically stated to act as antibody peptidyl-prolyl isomerase [2], did not show similar specificity to plasma cells (Appendix A). 

### 3.4. Expression of FKBP11 Is Induced by the Transcription Factor X-Box Binding Protein 1 (XBP1) and Protects an Alveolar Cell Line from ER-Stress Induced Cell Death

Plasma cell differentiation requires the activation of the unfolded protein response (UPR) [65], an ER stress-induced signaling pathway, and FKBP11 has been described as an ER-resident peptidyl prolyl isomerase [59]. We therefore assessed, using treatment with the ER stress inducer tunicamycin and subcellular fractionation, whether FKBP11 localized to the ER and was upregulated by ER stress in the B lymphocyte cell line Raji (derived from Burkitt’s lymphoma) and in A549 cells (derived from lung adenocarcinoma). In both cell types, the subcellular fractionation pattern of FKBP11 protein was consistent with ER residence, as judged by predominance in the microsomal fraction and colocalization with PDIA3 (Appendix A, see also Appendix A in phLF). The ER stress inducer tunicamycin upregulated *FKBP11* expression in a dose-dependent manner, in parallel to the well-established UPR target protein HSPA5 (A549, Figure 4A,B; Raji, Appendix A). In A549, a tunicamycin concentration of 0.1 µg/mL was sufficient to reproducibly induce the UPR (Figure 4A,B). Therefore, subsequent experiments for regulation of *FKBP11* expression in A549 were conducted in presence of 0.1 µg/mL tunicamycin to induce ER stress. The transcription factor X-box binding protein 1 (XBP1), a critical regulator of the UPR which governs late events of plasma cell differentiation [7,8,66], has been described to regulate expression of a subset of protein folding catalysts in this context [67]. Hence, we assessed whether also *FKBP11* is regulated in an XBP1-dependent manner. Indeed, siRNA-mediated knockdown of XBP1 in A549 cells after induction of ER stress by tunicamycin treatment, resulted in significant downregulation of *FKBP11* expression on transcript and protein level (Figure 4C,D). In agreement with a protective role under conditions of ER stress, knockdown of FKBP11 in A549 resulted in a higher susceptibility to tunicamycin-induced cell death (Figure 4E,F).

### 3.5. Recombinant FKBP11 Folds IgG Antibody In Vitro

Given ER residence and expression of *FKBP11* in antibody-producing plasma cells, we hypothesized that FKBP11 functions as an antibody foldase. To address this hypothesis, we took advantage of an in vitro antibody refolding assay, where, after full denaturation of IgG with guanidinium chloride, refolding kinetics in absence and presence of the purified recombinant FKBP domain of human FKBP11 was monitored by an IgG ELISA. As antigen-antibody binding strictly relies on the native three-dimensional structure, ELISA readouts are a measure of correctly folded IgG [14,57]. Recombinant PPIB, commonly accepted as antibody peptidyl-prolyl isomerase (PPIase) [2], was used as positive control. Addition of the recombinant FKBP domain of FKBP11 (amino acids Gly28 - Ala146 as described in Ishikawa et al. [59], for FKBP11 domain structure see Figure 5A) to denatured IgG resulted in an increase of both rate of refolding and total yield of refolded IgG as compared to negative controls (RNase and PBS, Figure 5B). This effect was inhibited by tacrolimus (FK506), a known inhibitor of many FKBPs, which binds to the PPIase activity-bearing FKBP domain [68] (Figure 5C). In comparison to the established antibody PPIase PPIB, considerably higher concentrations of FKBP11 (45 µM FKBP11 in comparison to 5 µM PPIB, see Figure 5B) were needed to demonstrate similar refolding activity. 

### 3.6. Neither Knockdown of FKBP11 Nor Knockdown of Cyclophilin B Affects IgG Yield of an Antibody-Producing Hybridoma Cell Line

The fusion of splenocytes from immunized mice or rats with a mouse myeloma cell line, the hybridoma technology, is a well-established method to generate cell lines for reproducible production of specific monoclonal antibodies, also at large-scale. In efforts to screen for a suitable hybridoma cell line for loss-of-function experiments, we first determined levels of FKBP11 and rat IgG in three hybridoma cell lines created by fusion of the mouse myeloma cell line P3X63-Ag8.653 (AG8) with splenocytes of differently immunized rats (H1-H3, Figure 6A). We verified expression of *Fkbp11* in all of these cell lines, but observed the highest protein levels of FKBP11 and rat IgG in the hybridoma cell line H3. We therefore chose H3 for loss-of-function experiments. ER residence of FKBP11 in H3 was confirmed by colocalization with concanavalin A using confocal microscopy (Figure 6B) and by subcellular fractionation, where FKBP11 showed the same enrichment pattern as the ER marker calreticulin (CALR, Figure 6C). While the subcellular fractionation results also indicated a nuclear localization of FKBP11, this could not be confirmed by confocal microscopy (no colocalization with DAPI in Figure 6B). Therefore, enrichment of FKBP11 and CALR in the nuclear fraction probably represent some contamination of the nuclear fraction by ER-resident proteins.

Delivery of *Fkbp11*- and *Ppib*-specific siRNA into H3 by electroporation resulted in acceptable knockdown efficiencies (Figure 6E), without affecting cell viability in comparison to non-targeting scrambled siRNA (scr, Figure 6F). Neither *Fkbp11*- nor *Ppib*-specific siRNA affected levels of intracellular IgG heavy (HC) or light chain (LC, Figure 6D, quantified in Figure 6G). Also levels of the ER stress marker HSPA5 were not altered by reductions in *Fkbp11* or *Ppib* expression (Figure 6D, quantified in Figure 6G). While we observed a small non-significant trend for decreased transcript for IgG HC in response to PPIB knockdown, the same remained unchanged in response to FKBP11 knockdown (Appendix A). Furthermore, knockdown of FKBP11 did not alter the subcellular fractionation pattern of IgG chains, which could have indicated accumulation of misfolded protein in the ER (Appendix A). Western Blot analysis of the hybridoma supernatants suggested a weak reduction of secreted IgG antibody chains for FKBP11 knockdown which, however, failed to reach significance (LC, *p* = 0.1203; HC, *p* = 0.2650; paired *t*-test; Figure 6H, quantified in Figure 6G, secreted LC/HC). For a more rigorous quantification of IgG in the hybridoma supernatants, we compared IgG levels after transfection of hybridoma cells with control, FKBP11 and PPIB siRNA by ELISA using serial dilutions of the hybridoma supernatants. Here, we observed no differences of IgG concentrations relative to control siRNA (Figure 6I). Finally, using a similar ELISA approach, we also assessed whether knockdown of FKBP11 or PPIB affected binding of secreted IgG to the cognate antigen and again found that there was no difference relative to control siRNA (Appendix A).

## 4. Discussion

A profound understanding of mechanisms of plasma cell differentiation and antibody folding is beneficial not only for the development and production of antibody-based therapeutics but also for targeting autoimmune diseases. In the current study, we identified the immunophilin FKBP11 as a novel plasma cell-specific antibody folding catalyst. Expression of *FKBP11* was increased in lungs with IPF, a fatal fibrotic disease with autoimmune features [43,44,45], and localized specifically to tissue-resident plasma cells in IPF lung as well as primary and secondary lymphatic organs. *FKBP11* was upregulated upon differentiation of B cells into antibody-secreting plasma cells and upon induction of ER stress in an XBP1-dependent manner in vitro. The purified FKBP domain of human FKBP11, which carries the PPIase activity, catalysed antibody refolding. While FKBP11 knockdown resulted in higher susceptibility to ER stress-induced cell death in A549 cells, viability and antibody yield of a hybridoma cell line was not affected relative to control siRNA.

Antibody function is strictly dependent on correct folding of the three-dimensional structure and peptidyl-prolyl isomerization is the rate-limiting step in the process of IgG folding [2]. However, the knowledge on PPIases catalysing this step is much based on circumstantial evidence and PPIases in the human plasma cell have not been well-described. In our study, we provide strong in vivo and in vitro evidence that human FKBP11 is a novel plasma cell-specific antibody PPIase. To date, two other PPIases have been proposed in the context of antibody folding, namely FKBP1A (FKBP12) and PPIB. Lilie et al. [14] provided the first conceptual evidence that immunophilins of the FKBP family may contribute to antibody folding, showing that purified FKBP1A acts as antibody PPIase in vitro. The authors used the recombinant protein and an IgG Fab fragment refolding assay [14], very similar to the one that we used (Figure 5B,C). However, FKBP1A is a cytosolic protein [15] and therefore unlikely to contribute to antibody folding in the ER in vivo.

PPIB is commonly accepted as antibody PPIase [2], a concept which, however, is still based on relatively few key indications: First, purified PPIB has been shown by Feige et al. [11] to catalyse immunoglobulin folding in vitro, a finding which we confirm in our study (Figure 5B). Second, using a chemical crosslinking approach in mouse lymphoma cell lines followed by mass spectrometry-based identification, PPIB was found to reside in a complex associated with unassembled, incompletely folded immunoglobulin heavy chains [10]. More recently, using a combination of ER-specific pull-down and a yeast-two hybrid system, Jansen et al. were able to map multiple interactions between ER foldases, and identified a novel complex between PPIB and the oxidoreductase ERp72 (encoded by *PDIA4*) [13]. In a folding assay which monitored disulphide bond formation between the constant parts of the IgG heavy and light chain, a C_H_1-C_L_ assembly assay [11], the authors could show that PPIB potentiates disulphide isomerase activity of Erp72 [13]. However, PPIB is a general ER-resident folding catalyst, for instance with an established role in collagen triple helix formation [69]; indeed sc-RNA-Seq data from mouse lungs (Appendix A) confirms that PPIB is abundantly expressed in almost all cell types. In the present study, by contrast, we not only show that FKBP11 directly catalyses antibody folding, but provide multiple evidence for plasma cell-specific upregulation in patient material and human test systems and demonstrate that *FKBP11* expression is induced by the UPR in an XBP1-dependent manner, a pathway crucial for late events in the course of B cell differentiation to antibody-producing plasma cells. Notably, our findings are backed up by previous transcriptomic studies on UPR-induced gene expression in the lymphoma cell line Raji and in fibroblasts, which also show upregulation of FKBP11 transcript [70,71]. In addition, we confirm ER residence of FKBP11 [72] and show that FKBP11 colocalizes in plasma cells with at least two major immunoglobulin classes, namely IgG and IgA. Our observations thus support a critical and specific function of FKBP11 in general plasma cell biology. In agreement, in work using a lentiviral transgenic mouse model, Ruer-Laventie et al. have provided evidence that *FKBP11* overexpression initiates plasma cell differentiation and results in higher serum levels of basal IgG3 [27]. Also, *FKBP11* expression in peripheral B cells from Systemic Lupus Erythematosus (SLE) patients, a severe and prototypic autoimmune disease, has been found to correlate with increased numbers of peripheral plasmablasts [27].

While our results of the IgG refolding assay establish FKBP11 as a novel plasma cell antibody foldase, knocking down FKBP11 in a hybridoma cell line was not sufficient to affect antibody yield or functionality. Notably, in our hands, the same was true for the established antibody foldase PPIB. We believe that this observation does not contradict a function as antibody foldase for FKBP11, but may rather reflect considerable redundancy in the antibody folding capacity in hybridoma cell lines. Overall, this underlines the general importance of the UPR, inducing a plethora of chaperones and folding catalysts, which collectively protect unfolded proteins from aggregation. Furthermore, it is important to acknowledge that, while PPIases typically accelerate folding kinetics, their absence does not much affect overall yield of correctly folded antibody fragments after denaturation and refolding in vitro [11,73], indicating considerable, albeit slow, auto-catalysis. Hence, therapeutic application of our findings for treatment of autoimmune disease may require additional targets for the depletion of plasma cells, but, given the comparably high level of plasma cell specificity, FKBP11-directed drugs could be used to specifically target the ER-resident folding machinery in plasma cells. 

Our finding that PPIB knockdown was not sufficient to decrease antibody yield is in part contradictory to a previous study, which reported that knockdown of PPIB, with similar knockdown efficiency as shown here, in murine hybridoma and primary B cells impaired IgG synthesis [12]. Unfortunately, the authors did not assess secreted IgG in that assay and the robustness of the latter findings remains unclear as only representative results without statistical analysis are given. 

FKBP2 (also termed FKBP13) is another immunophilin that recently has received attention in the context of plasma cell biology and IPF. While a direct antibody folding activity has to date not been reported, findings reported by Jeong et al. suggest that FKBP2 is also regulated by XBP1, directly interacts with immunoglobulins in the ER, but targets them for proteasome-mediated degradation and thereby reduces the overall level of ER stress [74]. Similar to *PPIB*, however, expression of *FKBP2* is not restricted to plasma cells, at least not in the lung (Appendix A) [64,75]. FKBP2 was enriched in fibrotic regions in IPF lungs, while, in the bleomycin-induced mouse model of lung fibrosis, deficiency of FKBP2 aggravated fibrogenesis and impaired resolution of fibrosis [75]. Overall, these findings rather point towards a protective role of FKBP2 in the context of fibrosis.

As FKBP11 was exclusively expressed by CD3^−^/CD20^−^/CD38^+^/CD27^+^ plasma cells in IPF lung tissue, we quantified this cell population in plasma of IPF patients. Here, in contrast to other studies [27,44], we did not gate for CD19^+^ cells and normalize to the total B cell population, as it has been reported that a significant proportion of plasma cells lose CD19 expression [76,77]. Therefore a direct comparison of our results with the published ones [27,44] is not possible. With an average of around 0.3% of all live cells, the resulting population was very small and we did not observe an increase as compared to healthy control blood samples (Figure 2C). This suggested that the majority of plasma cells in IPF lungs did not derive from peripheral organs but differentiated from local B cells within the fibrotic lung tissue. 

Several studies have indicated the presence of circulating autoantibodies and increased levels of B-lymphocyte stimulator factor (BLyS) in plasma of IPF patients [44,78]. In a recent proteomic study, we could show that upregulation of the plasma cell-specific protein MZB1 is a common feature of fibrotic lung diseases including IPF and, notably, also FKBP11 was among the significantly upregulated proteins in this study [43]. Even if our study provides circumstantial support for the autoimmune hypothesis in IPF, it still remains unclear at present whether these observations represent epiphenomena or actually reflect disease-causing autoimmune mechanisms. Our correlations of *FKBP11* transcript abundance with lung function and transcript abundance of typical fibrotic markers in IPF lungs suggest that there is no strong direct relationship between the level of scarring and the presence of FKBP11^+^ plasma cells (Appendix A), but a time- and space-resolved analysis of interstitial scarring and lung-resident plasma cell differentiation would be necessary to draw robust conclusions in terms of cause or consequence. Auto-antibody-targeted treatments, including therapeutic plasma exchange, have been beneficial in acute exacerbations of IPF [79] and a clinical phase II study is currently ongoing to evaluate the potential therapeutic efficacy of rituximab, a CD20 antibody which specifically targets B lymphocytes (ClinicalTrials.gov Identifier NCT01969409), in IPF patients. Our study suggests that it may be beneficial to evaluate the potential of more plasma cell-specific treatments, e.g., by use of CD38-specific antibodies some of which are currently in clinical trials for treatment of multiple myeloma [80]. 

Our study has several limitations. First, we used a mouse/rat hybridoma cell line as model for antibody secretion and not human plasma cells or an antibody-secreting plasmacytoma cell line. This was primarily due to the well-known technical challenges of electroporation/nucleofection of B-, plasma and myeloma cells [81]. While we, after multiple rounds of optimization, succeeded to achieve an acceptable and consistent knockdown efficiency for both FKBP11 and PPIB in the hybridoma cell line, we failed doing so for the plasma cell myeloma cell line JK-6L. Unfortunately, while the yield of our plasma cell differentiation assay (Figure 3) was sufficient for subsequent FACS sorting and qRT-PCR analysis of the resulting subpopulations, it did not allow for testing downstream genetic manipulation assays and effects on antibody yield. We considered using a Crispr-Cas9 approach in the hybridoma cell line for a clean knockout of FKBP11, which ultimately may have led to more conclusive results on the role of FKBP11 in antibody synthesis and secretion. But as CrisprCas9 approaches are a challenging task in polyploid cells, we ultimately decided against that, given the polyploid nature of the hybridoma cell line [82]. As to in vivo models, the FKBP11 knockout mouse is, to the best of our knowledge, not yet commercially available. Future studies using more efficient genetic manipulation approaches, human cells, and animal models are needed to further elucidate the role of FKBP11 in plasma cells. 

Second, our study does not establish whether FKBP11 is protective from fibrosis or contributes to disease development. Our observation that knockdown of FKBP11 increases susceptibility to cell death in the alveolar epithelial cell line A549, raises an important concern when considering FKBP11 inhibition as a therapeutic target in IPF. ER stress-induced apoptosis of alveolar epithelial cells is believed to be a triggering profibrotic event in IPF [83,84,85] and scRNA-Seq data demonstrate—comparatively weak, but detectable-expression of *FKBP11* in type I and II alveolar epithelial cells (type I and II pneumocytes). Targeting FKBP11 may therefore increase susceptibility to ER stress in remaining alveolar type II cells, perpetuating disease progression rather than halting it.

Of note, there is increasing evidence of involvement of multiple FKBPs in human disease. While FKBP10 [61], FKBP11 (this study), and FKBP2 [75] have been reported in the context of IPF, others have demonstrated a role of FKBP11 in autoimmune disease [27]. Additional FKBPs have been put forward as potential drug targets in steroid hormone-associated cancer and psychotic disorders (FKBP4, 5, 8), and Alzheimer’s disease (FKBP1A, 4, 5) [15,72,86]. Overall, it is becoming increasingly clear, that a systematic and detailed analysis of FK506 and FK506 analogues and their relative contribution to FKBP inhibition is warranted for the development of multiple novel therapeutic agents including the development of agents that lack FKBP1A-binding and calcineurin-inhibiting T cell suppressive activity.

## 5. Conclusions

Using a combination of patient material, human cell culture systems, and an in vitro refolding assay, the present study has identified *FKBP11* as an XBP1-driven UPR gene, which is highly expressed by plasma cells and encodes a peptidyl prolyl isomerase, which folds antibodies in vitro. Neither knockdown of FKBP11 nor PPIB in an antibody-producing hybridoma cell line affected antibody yield and functionality, indicating considerable redundancy in antibody folding mechanisms. Finally, the results also strengthen the underappreciated concept of autoimmune features in IPF.

## Figures and Tables

**Figure 1 cells-11-01341-f001:**
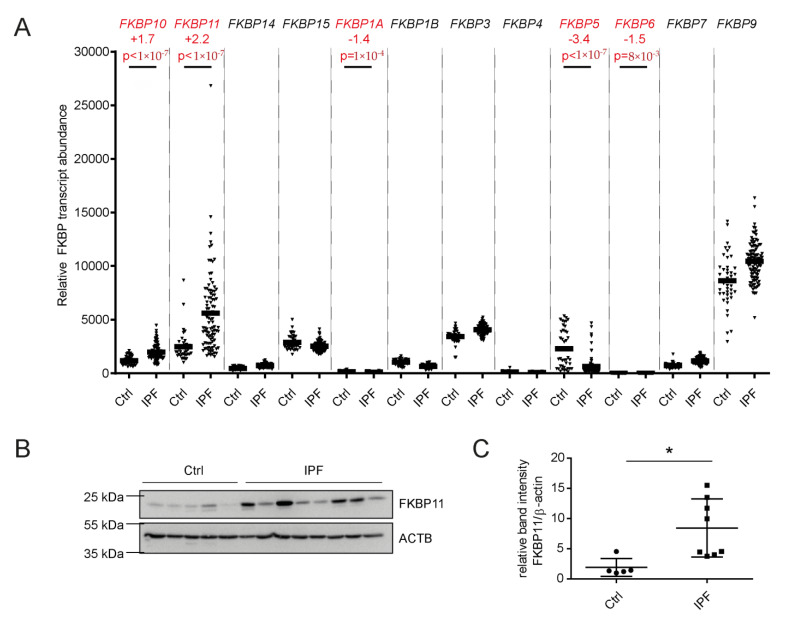
***FKBP11* is upregulated in idiopathic pulmonary fibrosis (IPF).** (**A**) Scatter plot for *FKBP* gene expression data extracted from microarray data of normal histology control (*n* = 43) and samples from patients with IPF (*n* = 99) [50,51]. (**B**) Western blot analysis of total lung tissue homogenate showed upregulation of FKBP11 in patients with IPF (*n* = 8) relative to donor samples (*n* = 5). (**C**) Densitometric analysis of the Western blot from (**B**). For Western Blot analysis, data shown are mean ± SEM, and a two-tailed Mann-Whitney test was used for statistical analysis (* *p* < 0.05). ACTB = β-actin as loading control.

**Figure 2 cells-11-01341-f002:**
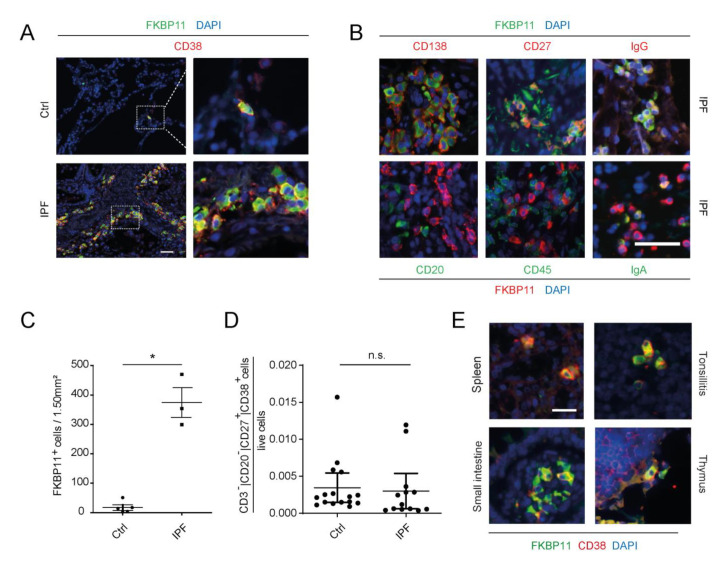
***FKBP11* is expressed by CD27^+^/CD38^+^/CD138^+^/CD20^−^/CD45^−^ plasma cells.** (**A**) Immunofluorescent stainings in control (Ctrl, upper panel) and IPF lung tissue sections (lower panel) demonstrate expression of FKBP11 in CD38^+^ plasma cells. Stainings are representative for *n* = 5 (Ctrl) and *n* = 3 (IPF). Scale bar 40 µm. (**B**) Immunofluorescent stainings in IPF lung tissue sections further demonstrate that FKBP11^+^ plasma cells are positive for CD138, CD27 (upper panel), but negative for CD20 and CD45 (lower panel) and produce mainly IgG (upper panel, far right), but also IgA (lower panel, far right). Note, that secondary antibody dyes for FKBP11 differ in upper and lower panel. Stainings of IPF lung sections are representative for *n* = 2 (CD138, CD27, CD20, CD45) and *n* = 3 (IgG, IgA). Scale bar, 40 µm. (**C**) Quantification of CD38/FKBP11 immunofluorescent stainings (**A**), based on lung sections from normal histology controls (*n* = 5) and IPF patients (*n* = 3, observer blinded to diagnosis), where FKBP11^+^/CD38^+^ cells from ten randomly selected images sized 1.5 mm^2^ were counted and added up for all 10 images (*p* = 0.0357, two-tailed Mann-Whitney (**D**) Numbers of circulating CD20^−^/CD27^+^/CD38^+^ plasma cells were not significantly changed (*p* = 0.2680, two-tailed Mann-Whitney test) between healthy subjects (*n* = 20) and IPF patients (*n* = 13). (**E**) Immunofluorescent stainings of a human tissue array demonstrated FKBP11^+^/CD38^+^ cells in other tissues than lung, namely spleen, tonsils, thymus, and small intestine. Scale bar 20 µm. * *p* < 0.05; n.s., not significant.

**Figure 3 cells-11-01341-f003:**
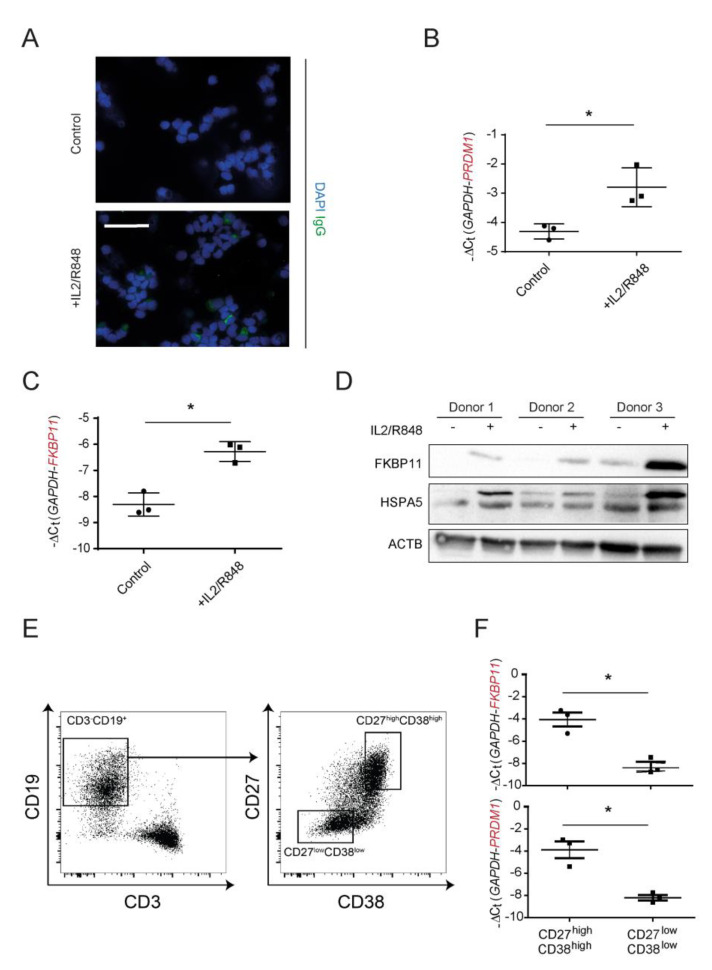
**In vitro plasma cell differentiation is accompanied by an increase of FKBP11 expression.** PBMCs from blood of three healthy volunteers were treated with interleukin-2 (IL2) and R848 to induce differentiation of memory B cells to antibody-producing plasma cells. (**A**) Increased IgG expression upon treatment was observed using immunofluorescence analysis of cytospins showing cells positive for intracellular IgG. Scale bar 50 µm. (**B**) IL2/R848 treatment led to increased levels of the B cell activation marker PR domain zinc finger protein 1 (PRDM1). (**C**,**D**) Expression of *FKBP11* at transcript (**C**) as well as protein (**D**) level was increased in the course of B cell to plasma cell differentiation. (**D**) Upregulation of the ER chaperone HSPA5 confirmed induction of the unfolded protein response (UPR) during B cell differentiation to plasma cells. (**E**) PBMCs of healthy donors were stimulated for 7 days with IL-2 and R848. Subsequently, the activated and differentiated cells were pre-gated on live and singlet cells and further gated on CD3^−^ and CD19^+^ B cells (left panel). Plasmablasts (CD19^+^CD27^high^CD38^high^) and non-plasmablast B cells (CD19^+^CD27^low^CD38^low^) were bulk sorted using FACSAria Fusion (right panel). Flow cytometry panels are displayed from one representative donor of three independent experiments. (**F**) Expression of *FKBP11* and *PRDM1* were analyzed by qPCR of the sorted cell fractions. ACTB = β-actin as loading control. For IgG concentrations and qPCR results, data shown are mean ± SEM, and a paired *t*-test was used for statistical analysis (* *p* ≤ 0.05; exact *p*-values: B: *p* = 0.0474; C: *p* = 0.0194; F top panel (FKBP11): *p* = 0.0180; F bottom panel (PRDM1): *p* = 0.0495).

**Figure 4 cells-11-01341-f004:**
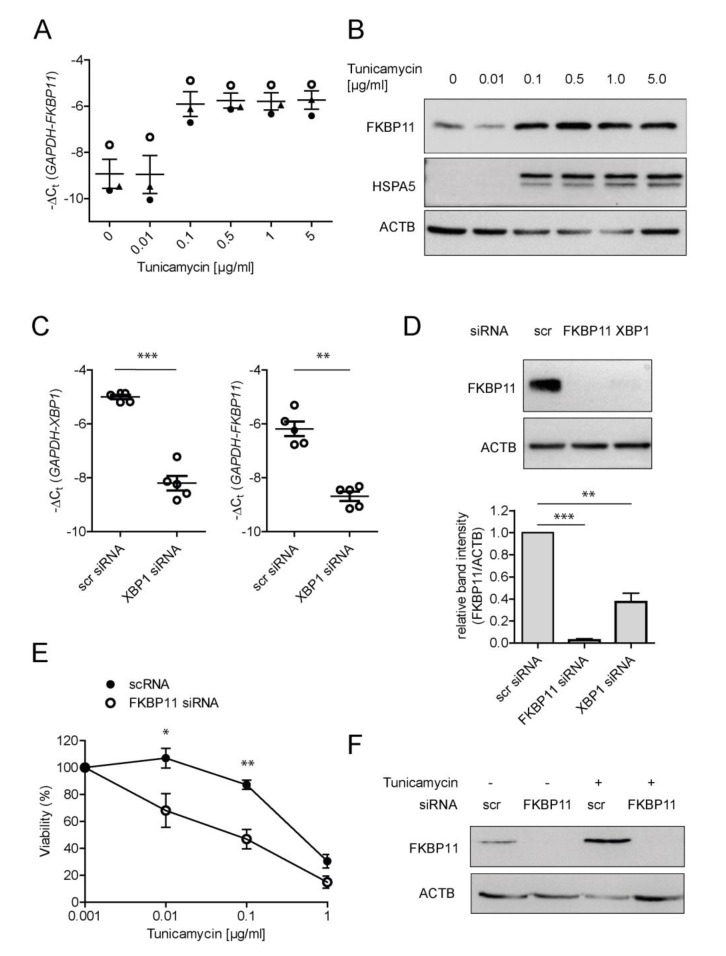
**FKBP11 is induced by ER stress in an XBP1-dependent manner and protects from ER-stress induced cell death.** (**A**,**B**) Treatment of A549 with the synthetic ER stress inducer tunicamycin led to a dose-dependent increase of *FKBP11* expression both on transcript (**A**) and on protein level (**B**). Upregulation of the ER chaperone HSPA5 confirmed induction of ER stress. ACTB = β-actin as loading control. (**C**) Knockdown of XBP1 in A549 cells treated with 0.1 µg/mL tunicamycin was efficient as assessed by qRT-PCR (left, *p* = 0.0007, paired *t*-test) and led to a drastic decrease of *FKBP11* transcript (right, *p* = 0.0026, paired *t*-test). (**D**) Knockdown of XBP1 equally led to loss of FKBP11 protein (*p* = 0.0014, paired *t*-test). This result is shown in comparison to FKBP11 knockdown under similar conditions (*p* < 0.0001, paired *t*-test); top panel, representative Western Blot; bottom panel corresponding densitometric analysis). (**E**) Combining FKBP11 knockdown with different concentrations of tunicamycin ranging from 1 ng/mL to 1 µg/mL and assessing cell viability by trypan blue exclusion showed that FKBP11 knockdown leads to higher susceptibility to ER stress-induced cell death (Exact *p*-values: 0.01 µg/mL: *p* = 0.0414; 0.1 µg/mL: *p* = 0.0068; paired *t*-test.) (**F**) Knockdown of FKBP11 in A549 cells was highly efficient in absence and presence of tunicamycin. For (**A**,**E**) data shown is based on three and four independent experiments, respectively, and given as mean ± SEM. (**B**,**F**) is representative for three independent experiments. For (**C**,**D**), data shown is based on five independent experiments and given as mean ± SEM. A paired *t*-test was used for statistical analysis (* *p* < 0.05; ** *p* < 0.01; *** *p* < 0.001).

**Figure 5 cells-11-01341-f005:**
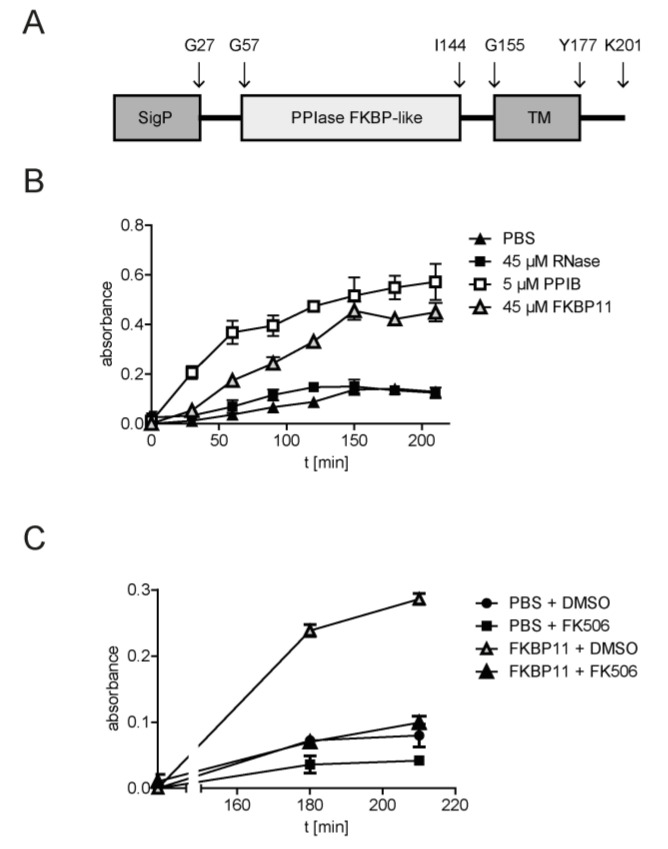
**Recombinant FKBP11 folds IgG antibody in vitro.** (**A**) Schematic representation of FKBP11 domain structure. For these experiments, the purified FKBP domain of FKBP11 (amino acids G28 - A146) without the N-terminal signal peptide (SigP) and the C-terminal transmembrane region (TM) was used. (**B**) In vitro antibody refolding kinetics in absence and presence of FKBP11 and recombinant PPIB as positive control. (**C**) Antibody refolding by FKBP11 was inhibited by tacrolimus (FK506). Data shown is based on three independent experiments and given as mean ± SEM; error bars are missing when they were smaller than the size of the data symbols.

**Figure 6 cells-11-01341-f006:**
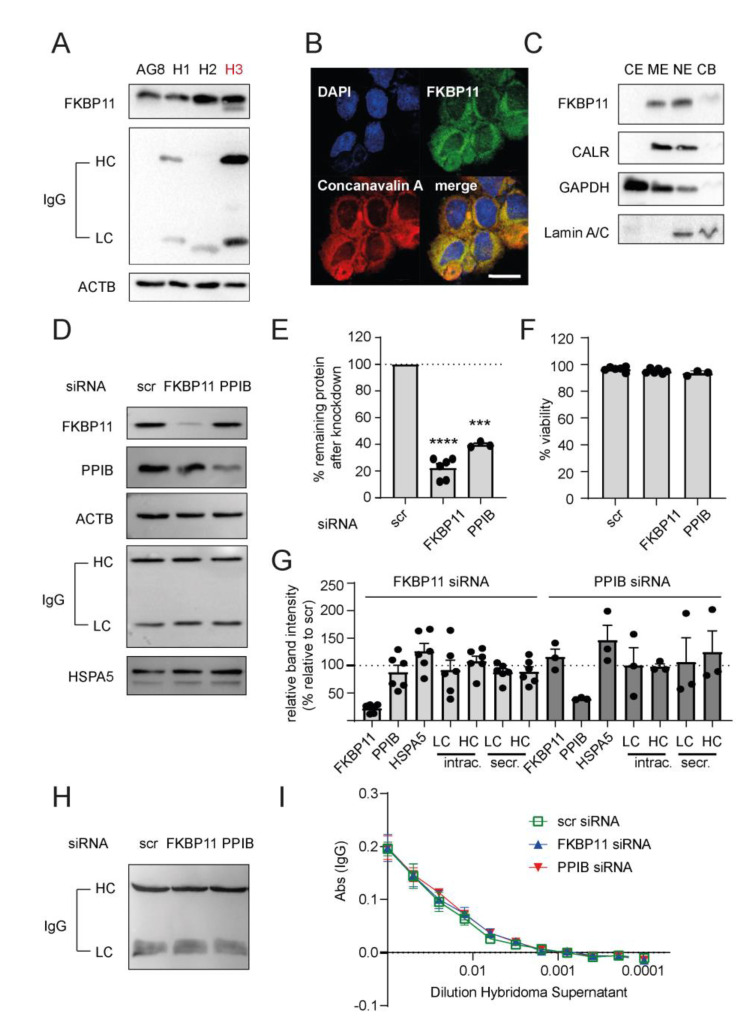
**Knockdown of FKBP11 or PPIB does not reduce antibody yield of a hybridoma cell line.** (**A**) Protein levels of FKBP11 and rat IgG in three different rat/mouse hybridoma cell lines (H1–H3) and the mouse myeloma cell line P3X63-Ag8.653 (AG8) used for fusion. H3 was chosen for subsequent experiments. (**B**) Confocal microscopy demonstrated colocalization of FKBP11 with the ER marker concanavalin A. Scale bar 20 µm. (**C**) Subcellular fractionation showed a similar enrichment pattern for FKBP11 as for the ER-resident protein calreticulin (CALR), with main localization in the microsomal (ME) and the nuclear extract (NE), but little to no detection in the cytosolic extract (CE) and the chromatin-bound fraction (CB). (**D**) Representative Western Blot analysis showing levels of FKBP11, the known antibody folding catalyst PPIB, loading control β-actin (ACTB), intracellular IgG antibody heavy (HC) and light chain (LC), and ER chaperone HSPA5, following siRNA-mediated knockdown of FKBP11 or PPIB in hybridoma cell line H3. (**E**) Mean FKBP11 knockdown efficiency in the rat hybridoma cell line H3 was 78 ± 3% (*n* = 6; paired *t*-test; **** *p* < 0.0001), mean PPIB knockdown efficiency in the same cell line was 60 ± 1% (*n* = 3; paired *t*-test; *** *p* < 0.001). (**F**) Knockdown of FKBP11 (*n* = 6) or PPIB (*n* = 3) did not affect cell viability relative to scr siRNA control. (**G**) Quantification of immunoblot band intensities (see representative immunoblots for intracellular protein in panel **D**, for secreted IgG in panel **H**) following FKBP11 or PPIB knockdown revealed no significant changes except for the siRNA targets FKBP11 and PPIB; data for the latter two is identical to panel E; intrac., intracellular; secr., secreted. (**H**) Representative Western Blot analysis showing levels of secreted IgG antibody heavy (HC) and light chain (LC), following siRNA-mediated knockdown of FKBP11 or PPIB in hybridoma cell line H3. Quantification is given in panel G (secr. IgG, LC, HC). (**I**) ELISA-based IgG quantification showed no significant effect on antibody secretion from FKBP11-, PPIB- or scr siRNA-transfected H3 cells (*n* = 3).

## Data Availability

Gene expression microarray data (Agilent Technologies, Santa Clara, CA) have been deposited under accession number GSE47460 in the data repository Gene Expression Omnibus (GEO, NCBI).

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
