# Peer review of "FK506-Binding Protein 11 Is a Novel Plasma Cell-Specific Antibody Folding Catalyst with Increased Expression in Idiopathic Pulmonary Fibrosis"

_cells, 2022, doi:10.3390/cells11081341_

Round 1

Reviewer 1 Report

I have read the article by Preisendörfer et al. with great interest. I would like to congratulate the authors for their elegant and comprehensive work. My only, but major remark is that I am not sure if IPF is a homogenous disease and I believe various processes could lead to the same clinical phenotype. Having said this, a marked variation of FKBP11 was noticed in the IPF population. I strongly believe that correlation analyses of FKBP11 expression with clinical variables/demographics is warranted to better understand this pathway.

Comments:

  • Was FKBP expression related to demographics and clinical parameters?
  • I understand that the clinical data can be find at differences sources than the manuscript. However, it would be essential to understand the clinical and demographic data of the patient and control group for Figure1 analyses, especially comorbidities in the IPF group as they might also influence the findings (as it was detailed in the background).
  • Figure 1. Please provide scatter dots rather than box and whiskers to better understand the distribution of data.
  • ELISA measurements. I presume they were performed in duplicates. Could you please provide the coefficient of variation of the results?

Author Response

Comment 1: Was FKBP expression related to demographics and clinical parameters. I understand that the clinical data can be find at differences sources than the manuscript. However, it would be essential to understand the clinical and demographic data of the patient and control group for Figure1 analyses, especially comorbidities in the IPF group as they might also influence the findings (as it was detailed in the background).

Answer 1: Thank you very much for this thoughtful comment. Indeed, IPF is a multifactorial and heterogeneous disease and believed to involve a complex interplay between environmental factors and genetic predisposition. The data we used is the publicly available gene expression dataset generated from the Lung Tissue Resource Consortium. This dataset was previously described at https://bmcgenomics.biomedcentral.com/articles/10.1186/s12864-015-2170-4 and has been extensively used (see review https://www.frontiersin.org/articles/10.3389/fmed.2018.00087/full). The diagnoses are based on the clinical guidelines for the time of the generation of the dataset. In this manuscript, to adhere with the reviewer’s request, we have now correlated FKBP11 gene expression with demographic and clinical data, namely age, sex, quality of life scores (SF-12, SGRQ), 6-minute walk test, smoking status, pack years, and lung function parameters like forced vital capacity (FVC) and diffusing capacity for carbon monoxide (DLCO). We include this data in correlation with FKBP11 in the revised version (Supplementary Figure S2). We did not observe any significant relationships except for a significant, but weak negative correlation with FVC (% of predicted) and a similar relationship with DLCO which, however, just failed significance. A detailed analysis of other potential comorbidities is beyond the scope of this article.

Nevertheless, inspired by this comment, we took the opportunity to use the available transcriptome data for correlation analysis between FKBP11, the plasma cell marker MZB1, and fibrotic markers like ACTA2 (alpha-SMA), COL1A2 (alpha2-chain of type I collagen, transcript data of COL1A1 was not available in the data bank), and COL3A1 (type III collagen) expression in those tissues. We observed a very strong (R=0.93) and highly significant (p=4.43e-43) correlation with MZB1, further confirming FKBP11 expression in plasma cells in IPF lung tissue. In contrast, FKBP11 did not or only weakly correlate with ACTA2, COL1A2, and COL3A1. Overall, this clearly underlines plasma cell specificity of FKBP11 expression in the fibrotic lung but suggests a limited association with interstitial scarring and the resulting loss of lung function. These findings are now shortly discussed in addition.

Comment 2: Figure 1. Please provide scatter dots rather than box and whiskers to better understand the distribution of data.

Answer 2: Thank you for that comment; we have altered the graph in Figure 1 accordingly.

Comment 3: ELISA measurements. I presume they were performed in duplicates. Could you please provide the coefficient of variation of the results?

Answer 3: Thank you for this comment. In general, we would like to point out that we do not perform classical ELISAs to obtain absolute concentrations where coefficients of variation would be most important, but use ELISA-based methods for antibody folding kinetics (Figure 5) and titrations of antibody-containing supernatants (Figure 6, Supplementary Figure S9). Therefore, our data needs to be interpreted in the context of all samples including time dependency, concentration-dependency and negative and positive controls rather than looked at as absolute values with a specific variation.

As we have these two different types of ELISA measurements in this manuscript, we are not entirely sure we understand which one the reviewer is referring to, but we assume it is Figure 5, because, in the graphs showing refolding experiments, several data points are given without an error bar, looking as if they corresponded to single data points only. However, all these results are indeed given as mean +/- SEM of n=3 independent experiments; the error bars were not plotted by GraphPad prism for some values (e.g. control PBS, RNase), simply because they were shorter than the size of the data symbol. We considered showing all three replicates in the graph for each curve but that was not helpful for the data where error bars were missing and just made the graph much less reader-friendly. Instead, we now state in the figure legend that missing error bars were smaller than the data symbols and hope that we satisfactorily answered this reviewer’s question.

Reviewer 2 Report

Overall, this manuscript is a good paper, well conducted and organized, and by using samples from IPF patients, invitro models, as well as cutting-edge technologies, the evidence places FKBP11 as a novel antibody folding catalyst that is stimulated by the UPR in an XBP1-dependent manner; moreover, the authors show that FKBP11 is preferentially produced by human plasma cells and it is overexpressed in IPF. However, some minor issues still need to be addressed before further steps.

Include the symbol of X-box-binding protein 1 at the first appearance in the abstract section, as follows: Induction of ER stress in cell lines demonstrated induction of FKBP11 in the context of the unfolded protein response in an X-box-binding protein 1 (XBP1)-dependent manner.

Based on HUGO Gene Nomenclature Committee, gene and protein symbols must be indicated according to the accepted nomenclatures. So, throughout the manuscript including abstract, main text, figures, figures legends, and supplementary information gene and protein symbols must be properly indicated. Authors should know that both gene and protein symbols are different among species. As reference, authors should review HUGO website as well as the following article: PMID: 22836666.

NCBI Accession Numbers of evaluated genes should be included in supplementary table S1.

Homogenize the “n” citation throughout the manuscript, either like this (n = 99) or like this (n=8), it is preferable with space in between; moreover, letter “n” should be in italic. Same suggestion for “p” value citation.

Much of the information in the legend of figure 1 is repetitive in the result section: “FKBP11 expression is increased in IPF lungs”. In general, most of figure legends contain repetitive information.

Discussion section contains repetitive information from result section, as a result, the manuscript is too long. So, it can be easily shortened by a 30% approx.

It is not necessary to capitalize the word “NOVEL” in the title. As a general rule for the publication of scientific findings, reports must contain novel discoveries, otherwise they would not be published by any journal.

Author Response

Comment 1: Include the symbol of X-box-binding protein 1 at the first appearance in the abstract section, as follows: Induction of ER stress in cell lines demonstrated induction of FKBP11 in the context of the unfolded protein response in an X-box-binding protein 1 (XBP1)-dependent manner.

Answer 1: Thank you for this observant comment – we corrected this in the revised version.

Comment 2: Based on HUGO Gene Nomenclature Committee, gene and protein symbols must be indicated according to the accepted nomenclatures. So, throughout the manuscript including abstract, main text, figures, figures legends, and supplementary information gene and protein symbols must be properly indicated. Authors should know that both gene and protein symbols are different among species. As reference, authors should review HUGO website as well as the following article: PMID: 22836666.

Answer 2: Thank you for this valuable comment and for bringing our attention to this truly entertaining article! We have now carefully revised our article to cohere with the following guidelines: (1) As protein name, we now use, except for one exception mentioned below, the same as the approved gene symbol, gene and transcript in italics, protein not italiced. (2) For mouse and rat genes and transcripts, we use italicized names with only one initial capital letter while for proteins, we use non-italicized names, all in capital letters. We have also changed all of this consistently in the figures and in the supplement. For knockdowns, however, we kept protein names, as we mostly validated these on protein level. Also, for Lamin A/C (Figure 6, Supplementary Figures S3 and S8), we kept this protein name recommended by UniProt which refers to the processed form of the protein which we detect on the Western Blot, because using the gene name as protein name (LMNA) would suggest we detect the unprocessed form. We hope that with this, we have satisfactorily addressed this reviewer’s comment.

Comment 3: NCBI Accession Numbers of evaluated genes should be included in supplementary table S1.

Answer 3: Thank you for this comment, we have now included all relevant NCBI accession numbers in that table. When we updated that table we furthermore changed “BLIMP1” to the correct gene name PRDM1.

Comment 4: Homogenize the “n” citation throughout the manuscript, either like this (n = 99) or like this (n=8), it is preferable with space in between; moreover, letter “n” should be in italic. Same suggestion for “p” value citation.

Answer 4: Thank you for this comment. As recommended by the reviewer, we have homogenized this for both “n” and “p” and consistently used italics and space in between.

Comment 5: Much of the information in the legend of figure 1 is repetitive in the result section: “FKBP11 expression is increased in IPF lungs”. In general, most of figure legends contain repetitive information.

Answer 5: Thank you for this comment. We agree and have removed much of the repetitive information from either figure legends (for the most part) ore results section.

Comment 6: Discussion section contains repetitive information from result section, as a result, the manuscript is too long. So, it can be easily shortened by a 30% approx.

Answer 6: We thank the reviewer for this very valuable comment. We have carefully revised the discussion accordingly and removed some repetitive information including also referrals to figures that probably strengthened the impression of repetition. Nevertheless, with all respect for the reviewer, we think that some repetition is needed to discuss our findings in the context of the literature and for this, we think it is convenient for the reader to shortly mention key findings again. However, we believe that we have limited these instances to the necessary minimum now and, together with shortening of the figure legends, made the manuscript much more concise.

Comment 7: It is not necessary to capitalize the word “NOVEL” in the title. As a general rule for the publication of scientific findings, reports must contain novel discoveries, otherwise they would not be published by any journal.

Answer 7: Unfortunately, the word “novel” was capitalized when the manuscript portal automatically formatted our manuscript to the Cells’ template. We had capitalized the whole title in our original submission – we may have inconsistently used “capital letter formatting” in the title leading to this error after transfer to the Cells template.

Round 2

Reviewer 1 Report

I am happy with the changes and suggest acceptance